# Modulating early visual processing by language

**Harm de Vries**[*]
University of Montreal
mail@harmdevries.com

**Florian Strub**[*]
Univ. Lille, CNRS, Centrale Lille,
Inria, UMR 9189 CRIStAL
florian.strub@inria.fr

**Jérémie Mary**[†]
Univ. Lille, CNRS, Centrale Lille,
Inria, UMR 9189 CRIStAL
jeremie.mary@univ-lille3.fr

**Hugo Larochelle**
Google Brain
hugolarochelle@google.com

**Olivier Pietquin**
DeepMind
pietquin@google.com

**Aaron Courville**
University of Montreal
aaron.courville@gmail.com

## Abstract

It is commonly assumed that language refers to high-level visual concepts while leaving low-level visual processing unaffected. This view dominates the current literature in computational models for language-vision tasks, where visual and linguistic inputs are mostly processed independently before being fused into a single representation. In this paper, we deviate from this classic pipeline and propose to modulate the *entire visual processing* by a linguistic input. Specifically, we introduce Conditional Batch Normalization (CBN) as an efficient mechanism to modulate convolutional feature maps by a linguistic embedding. We apply CBN to a pre-trained Residual Network (ResNet), leading to the MODulatEd ResNet (MODERN) architecture, and show that this significantly improves strong baselines on two visual question answering tasks. Our ablation study confirms that modulating from the early stages of the visual processing is beneficial.

## 1 Introduction

Human beings combine the processing of language and vision with apparent ease. For example, we can use natural language to describe perceived objects and we are able to imagine a visual scene from a given textual description. Developing intelligent machines with such impressive capabilities remains a long-standing research challenge with many practical applications.

Towards this grand goal, we have witnessed an increased interest in tasks at the intersection of computer vision and natural language processing. In particular, image captioning [16], visual question answering (VQA)[1, 23] and visually grounded dialogue systems[5, 6] constitute a popular set of example tasks for which large-scale datasets are now available. Developing computational models for language-vision tasks is challenging, especially because of the open question underlying all these tasks: how to fuse/integrate visual and textual representations? To what extent should we process visual and linguistic input separately, and at which stage should we fuse them? And equally important, what fusion mechanism to use?

In this paper, we restrict our attention to the domain of visual question answering which is a natural testbed for fusing language and vision. The VQA task concerns answering open-ended questions about images and has received significant attention from the research community [1, 9, 17, 23]. Current state-of-the-art systems often use the following computational pipeline [2, 17, 20] illustrated in Fig 1. They first extract *high-level* image features from an ImageNet pretrained convolutional network (e.g. the activations from a ResNet network [12]), and obtain a language embedding using a

---

[*]The first two authors contributed equally
[†]Now at Criteo

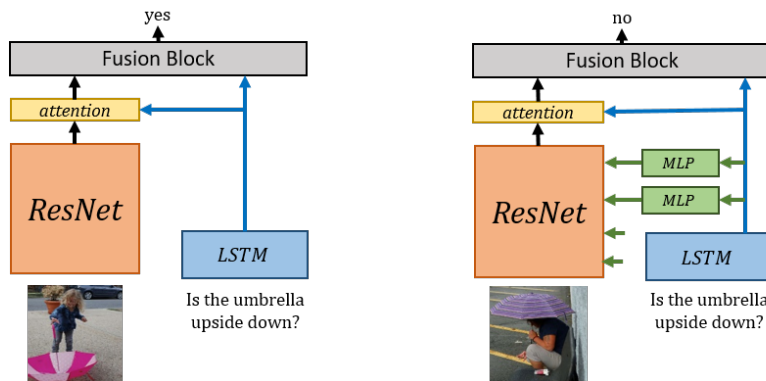

Figure 1: An overview of the classic VQA pipeline (left) vs ours (right). While language and vision modalities are independently processed in the classic pipeline, we propose to directly modulate ResNet processing by language.

recurrent neural network (RNN) over word-embeddings. These two high-level representations are then fused by concatenation [17], element-wise product [11, 13, 14, 17], Tucker decomposition [2] or compact bilinear pooling [9], and further processed for the downstream task at hand. Attention mechanisms [27] are often used to have questions attend to specific spatial locations of the extracted higher-level feature maps.

There are two main reasons for why the recent literature has focused on processing each modality independently. First, using a pretrained convnet as feature extractor prevents overfitting; Despite a large training set of a few hundred thousand samples, backpropagating the error of the downstream task into the weights of all layers often leads to overfitting. Second, the approach aligns with the dominant view that language interacts with high-level visual concepts. Words, in this view, can be thought of as "pointers" to high-level conceptual representations. To the best of our knowledge, this work is the first to fuse modalities at the very early stages of the image processing.

In parallel, the neuroscience community has been exploring to what extent the processing of language and vision is coupled [8]. More and more evidence accumulates that words set visual priors which alter how visual information is processed from the very beginning [3, 15, 24]. More precisely, it is observed that P1 signals, which are related to low-level visual features, are modulated while hearing specific words [3]. The language cue that people hear ahead of an image activates visual predictions and speed up the image recognition process. These findings suggest that independently processing visual and linguistic features might be suboptimal, and fusing them at the early stage may help the image processing.

In this paper, we introduce a novel approach to have language modulate the *entire* visual processing of a pre-trained convnet. We propose to condition the batch normalization [21] parameters on linguistic input (e.g., a question in a VQA task). Our approach, called Conditional Batch Normalization (CBN), is inspired by recent work in style transfer [7]. The key benefit of CBN is that it scales linearly with the number of feature maps in a convnet, which impacts less than 1% of the parameters, greatly reducing the risk of over-fitting. We apply CBN to a pretrained Residual Network, leading to a novel architecture to which we refer as MODERN. We show significant improvements on two VQA datasets, VQAv1 [1] and GuessWhat?! [6], but stress that our approach is a general fusing mechanism that can be applied to other multi-modal tasks.

To summarize, our contributions are three fold:

- We propose conditional batch normalization to modulate the entire visual processing by language from the early processing stages,

- We condition the batch normalization parameters of a pretrained ResNet on linguistic input, leading to a new network architecture: MODERN,

- We demonstrate improvements on state-of-the-art models for two VQA tasks and show the contribution of this modulation on the early stages.

## 2  Background

In this section we provide preliminaries on several components of our proposed VQA model.

### 2.1  Residual networks

We briefly outline residual networks (ResNets) [12], one of the current top-performing convolutional networks that won the ILSVRC 2015 classification competition. In contrast to precursor convnets (e.g. VGG[22]) that constructs a new representation at each layer, ResNet iteratively refines a representation by adding residuals. This modification enables to train very deep convolutional networks without suffering as much from the vanishing gradient problem. More specifically, ResNets are built from residual blocks:

$$F^{k+1} = \text{ReLU}(F^k + R(F^k)) \tag{1}$$

where $F^k$ denotes the outputted feature map. We will refer to $F_{i,c,w,h}$ to denote the $i^{\text{th}}$ input sample of the $c^{\text{th}}$ feature map at location $(w, h)$. The residual function $R(F^k)$ is composed of three convolutional layers (with a kernel size of 1, 3 and 1, respectively). See Fig. 2 in the original ResNet paper [12] for a detailed overview of a residual block.

A group of blocks is stacked to form a *stage* of computation in which the representation dimensionality stays identical. The general ResNet architecture starts with a single convolutional layer followed by four stages of computation. The transition from one stage to another is achieved through a projection layer that halves the spatial dimensions and doubles the number of feature maps. There are several pretrained ResNets available, including ResNet-50, ResNet-101 and ResNet-152 that differ in the number of residual blocks per stage.

### 2.2  Batch Normalization

The convolutional layers in ResNets make use of Batch Normalization (BN), a technique that was originally designed to accelarate the training of neural networks by reducing the internal co-variate shift [21]. Given a mini-batch $\mathcal{B} = \{F_{i,\cdot,\cdot,\cdot}\}_{i=1}^N$ of $N$ examples, BN normalizes the feature maps at training time as follows:

$$BN(F_{i,c,h,w}|\gamma_c, \beta_c) = \gamma_c \frac{F_{i,c,w,h} - \text{E}_\mathcal{B}[F_{\cdot,c,\cdot,\cdot}]}{\sqrt{\text{Var}_\mathcal{B}[F_{\cdot,c,\cdot,\cdot}] + \epsilon}} + \beta_c, \tag{2}$$

where $\epsilon$ is a constant damping factor for numerical stability, and $\gamma_c$ and $\beta_c$ are trainable scalars introduced to keep the representational power of the original network. Note that for convolutional layers the mean and variance are computed over both the batch and spatial dimensions (such that each location in the feature map is normalized in the same way). After the BN module, the output is fed to a non-linear activation function. At inference time, the batch mean $\text{E}_\mathcal{B}$ and variance $\text{Var}_\mathcal{B}$ are replaced by the population mean $\mu$ and variance $\sigma^2$, often estimated by an exponential moving average over batch mean and variance during training.

### 2.3  Language embeddings

We briefly recap the most common way to obtain a language embedding from a natural language question. Formally, a question $q = [w_k]_{k=1}^K$ is a sequence of length $K$ with each token $w_k$ taken from a predefined vocabulary $V$. We transform each token into a dense word-embedding $e(w_k)$ by a learned look-up table. For task with limited linguistic corpora (like VQA), it is common to concatenate pretrained Glove[19] vectors to the word embeddings. The sequence of embeddings $[e(w_k)]_{k=1}^K$ is then fed to a recurrent neural network (RNN), which produces a sequence of RNN state vectors $[\boldsymbol{s}_k]_{k=1}^K$ by repeatedly applying the transition function $f$:

$$\boldsymbol{s}_{k+1} = f(\boldsymbol{s}_k, e(w_k)). \tag{3}$$

Popular transition functions, like a long-short term memory (LSTM) cell [10] and a Gated Recurrent Unit (GRU)[4], incorporate gating mechanisms to better handle long-term dependencies. In this work, we will use an LSTM cell as our transition function. Finally, we take the last hidden state $s_I$ as the embedding of the question, which we denote as $\boldsymbol{e_q}$ throughout the rest of this paper.

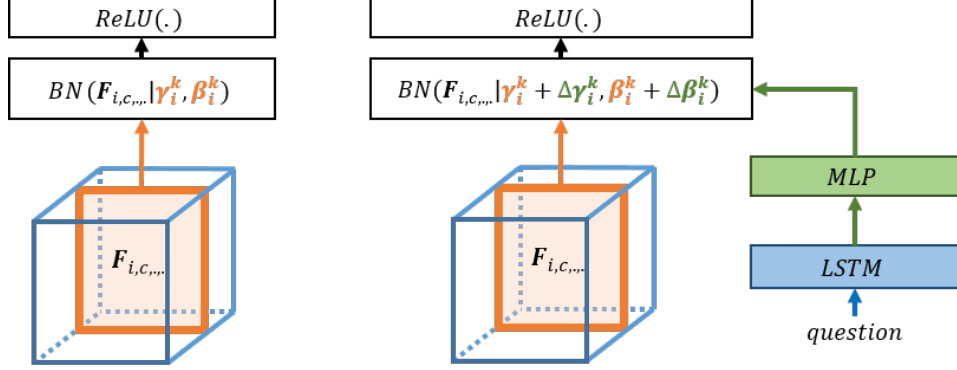

Figure 2: An overview of the computation graph of batch normalization (left) and conditional batch normalization (right). Best viewed in color.

## 3 Modulated Residual Networks

In this section we introduce conditional batch normalization, and show how we can use it to modulate a pretrained ResNet. The key idea is to predict the $\gamma$ and $\beta$ of the batch normalization from a language embedding. We first focus on a single convolutional layer with batch normalization module $BN(F_{i,c,h,w}|\gamma_c, \beta_c)$ for which pretrained scalars $\gamma_c$ and $\beta_c$ are available. We would like to directly predict these affine scaling parameters from our language embedding $e_q$. When starting the training procedure, these parameters must be close to the pretrained values to recover the original ResNet model as a poor initialization could significantly deteriorate performance. Unfortunately, it is difficult to initialize a network to output the pretrained $\gamma$ and $\beta$. For these reasons, we propose to predict a change $\Delta\beta_c$ and $\Delta\gamma_c$ on the frozen original scalars, for which it is straightforward to initialize a neural network to produce an output with zero-mean and small variance.

We use a one-hidden-layer MLP to predict these deltas from the question embedding $e_q$ for all feature maps within the layer:

$$\boldsymbol{\Delta\beta} = MLP(\boldsymbol{e_q}) \qquad \boldsymbol{\Delta\gamma} = MLP(\boldsymbol{e_q}) \tag{4}$$

So, given a feature map with $C$ channels, these MLPs output a vector of size $C$. We then add these predictions to the $\beta$ and $\gamma$ parameters:

$$\hat{\beta}_c = \beta_c + \Delta\beta_c \qquad \hat{\gamma}_c = \gamma_c + \Delta\gamma_c \tag{5}$$

Finally, these updated $\hat{\beta}$ and $\hat{\gamma}$ are used as parameters for the batch normalization: $BN(F_{i,c,h,w}|\hat{\gamma}_c, \hat{\beta}_c)$. We stress that we freeze all ResNet parameters, including $\boldsymbol{\gamma}$ and $\boldsymbol{\beta}$, during training. In Fig. 2, we visualize the difference between the computational flow of the original batch normalization and our proposed modification. As explained in section 2.1, a ResNet consists of four stages of computation, each subdivided in several residual blocks. In each block, we apply CBN to the three convolutional layers, as highlighted in Fig. 3.

CBN is a computationally efficient and powerful method to modulate neural activations; It enables the linguistic embedding to manipulate entire feature maps by scaling them up or down, negating them, or shutting them off, etc. As there only two parameters per feature map, the total number of BN parameters comprise less than 1% of the total number of parameters of a pre-trained ResNet. This makes CBN a very scalable method compared to conditionally predicting the weight matrices (or a low-rank approximation to that).

## 4 Experimental setting

We evaluate the proposed conditional batch normalization on two VQA tasks. In the next section, we outline these tasks and describe the neural architectures we use for our experiments. The source code for our experiments is available at `https://github.com/GuessWhatGame`. The hyperparameters are also provided in Appendix A.

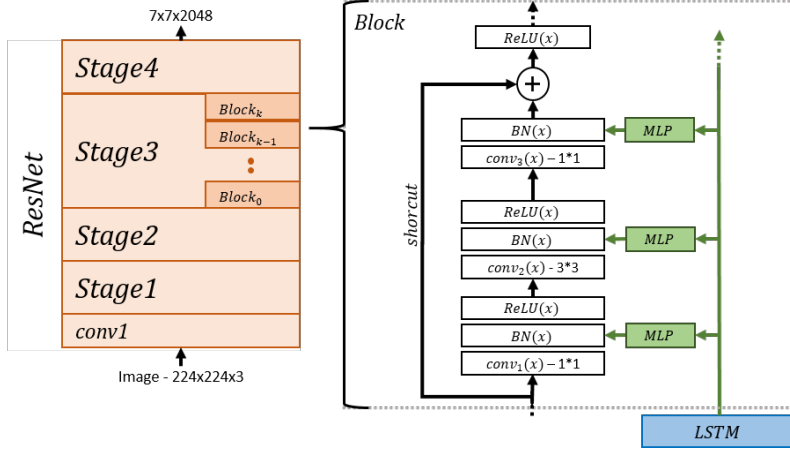

Figure 3: An overview of the MODERN architecture conditioned on the language embedding. MODERN modulates the batch norm parameters in all residual blocks.

## 4.1 VQA

The Visual Question Answering (VQA) task consists of open-ended questions about real images. Answering these questions requires an understanding of vision, language and commonsense knowledge. In this paper, we focus on VQAv1 dataset [1], which contains 614K questions on 204K images.

Our baseline architecture first obtains a question embedding $e_q$ by an LSTM-network, as further detailed in section 2.3. For the image, we extract the feature maps $F$ of the last layer of ResNet-50 (before the pooling layer). For input of size $224x224$ these feature maps are of size $7x7$, and we incorporate a spatial attention mechanism, conditioned on the question embedding $e_q$, to pool over the spatial dimensions. Formally, given a feature maps $F_{i,\cdot,\cdot,\cdot}$ and question embedding $e_q$, we obtain a visual embedding $e_v$ as follows:

$$\xi_{w,h} = MLP([\boldsymbol{F_{i,\cdot,w,h}}; \boldsymbol{e_q}]) \quad ; \quad \alpha_{w,h} = \frac{\exp(\xi_{w,h})}{\sum_{w,h}\exp(\xi_{w,h})} \quad ; \quad \boldsymbol{e_v} = \sum_{w,h}\alpha_{w,h}\boldsymbol{F_{i,\cdot,w,h}} \quad (6)$$

where $[\boldsymbol{F_{i,\cdot,w,h}}; \boldsymbol{e_q}]$ denotes concatenating the two vectors. We use an MLP with one hidden layer and ReLU activations whose parameters are shared along the spatial dimensions. The visual and question embedding are then fused by an element-wise product [1, 13, 14] as follows:

$$\text{fuse}(\boldsymbol{e_q}, \boldsymbol{e_v}) = \boldsymbol{P}^T\left((\tanh(\boldsymbol{U}^T\boldsymbol{e_q})) \circ (\tanh(\boldsymbol{V}^T\boldsymbol{e_v}))\right) + \boldsymbol{b}_P, \quad (7)$$

where $\circ$ denotes an element-wise product, and $\boldsymbol{P}$, $\boldsymbol{U}$ and $\boldsymbol{V}$ are trainable weight matrices and $\boldsymbol{b}_P$ is a trainable bias. The linguistic and perceptual representations are first projected to a space of equal dimensionality, after which a tanh non-linearity is applied. A fused vector is then computed by an element-wise product between the two representations. From this joined embedding we finally predict an answer distribution by a linear layer followed by a softmax activation function.

We will use the described architecture to study the impact CBN when using it in several stages of the ResNet. As our approach can be combined with any existing VQA architecture, we also apply MODERN to MRN [13, 14], a state-of-the-art network for VQA More specifically, this network replaces the classic attention mechanism with a more advanced one that included $g$ glimpses over the image features:

$$\xi_{w,h}^g = \boldsymbol{P}_{\alpha^g}^T(\tanh(\boldsymbol{U}'^T\boldsymbol{q}) \circ \tanh(\boldsymbol{V}'^T\boldsymbol{F_{i,\cdot,w,h}}^T))) \quad ; \quad \alpha_{w,h}^g = \frac{\exp(\xi_{w,h}^g)}{\sum_{w,h}\exp(\xi_{w,h}^g)} \quad (8)$$

$$\boldsymbol{e_v} = \Big\|_g \sum_{w,h}\alpha_{w,h}^g\boldsymbol{F_{i,\cdot,w,h}} \quad (9)$$

where $\boldsymbol{P}_{\alpha^g}$ is a trainable weight matrix defined for each glimpse $g$, $\boldsymbol{U}'$ and $\boldsymbol{V}'$ are trainable weight matrices shared among the glimpses and $\|$ concatenate vectors over their last dimension.

Table 1: VQA accuracies trained with train set and evaluated on test-dev.

| | Answer type | Yes/No | Number | Other | Overall |
|---|---|---|---|---|---|
| 224x224 | Baseline | 79.45% | 36.63% | 44.62% | 58.05% |
| | Ft Stage 4 | 78.37% | 34.27% | 43.72% | 56.91% |
| | Ft BN | 80.18% | 35.98% | 46.07% | 58.98% |
| | MODERN | 81.17% | 37.79% | 48.66% | 60.82% |
| 448x448 | MRN [14] with ResNet-50 | 80.20% | 37.73% | 49.53% | 60.84% |
| | MRN [14] with ResNet-152 | 80.95% | 38.39% | 50.59% | 61.73% |
| | MUTAN+MLB [2] | 82.29% | 37.27% | 48.23% | 61.02% |
| | MCB + Attention [9] with ResNet-50 | 60.46% | 38.29% | 48.68% | 60.46% |
| | MCB + Attention [9] with ResNet-152 | - | - | - | 62.50% |
| | MODERN | 81.38% | 36.06% | 51.64% | 62.16% |
| | MODERN + MRN [14] | 82.17% | 38.06% | 52.29% | **63.01**% |

Table 2: Ablation study to investigate the impact of leaving out the lower stages of ResNet.

(a) VQA, higher is better

| CBN applied to | Val. accuracy |
|---|---|
| $\emptyset$ | 56.12% |
| Stage 4 | 57.68% |
| Stages $3-4$ | 58.29% |
| Stages $2-4$ | 58.32% |
| All | **58.56**% |

(b) GuessWhat?!, lower is better

| CBN applied to | Test error |
|---|---|
| $\emptyset$ | 29.92% |
| Stage 4 | 26.42% |
| Stages $3-4$ | 25.24% |
| Stages $2-4$ | 25.31% |
| All | **25.06**% |

Noticeably, MODERN modulates the entire visual processing pipeline and therefore backpropagates through all convolutional layers. This requires much more GPU memory than using extracted features. To feasibly run such experiments on today's hardware, we conduct all experiments in this paper with a ResNet-50.

As for our training procedure, we select the 2k most-common answers from the training set, and use a cross-entropy loss over the distribution of provided answers. We train on the training set, do early-stopping on the validation set, and report the accuracies on the test-dev using the evaluation script provided by [1].

## 4.2 GuessWhat?!

GuessWhat?! is a cooperative two-player game in which both players see the image of a rich visual scene with several objects. One player – the Oracle – is randomly assigned an object in the scene. This object is not known by the other player – the questioner – whose goal it is to locate the hidden object by asking a series of yes-no questions which are answered by the Oracle [6].

The full dataset is composed of 822K binary question/answer pairs on 67K images. Interestingly, the GuessWhat?! game rules naturally leads to a rich variety of visually grounded questions. As opposed to the VQAv1 dataset, the dataset contains very few commonsense questions that can be answered without the image.

In this paper, we focus on the Oracle task, which is a form of visual question answering in which the answers are limited to yes, no and not applicable. Specifically, the oracle may take as an input the incoming question $q$, the image $I$ and the target object $o*$. This object can be described with its category $c$, its spatial location and the object crop.

We outline here the neural network architecture that was reported in the original GuessWhat?! paper [6]. First, we crop the initial image by using the target object bounding box object and rescale it to a 224 by 224 square. We then extract the activation of the last convolutional layer after the ReLU (stage4) of a pre-trained ResNet-50. We also embed the spatial information of the crop within the image by extracting an 8-dimensional vector of the location of the bounding box

$$[x_{min}, y_{min}, x_{max}, y_{max}, x_{center}, y_{center}, w_{box}, h_{box}], \tag{10}$$

Table 3: GuessWhat?! test errors for the Oracle model with different embeddings. Lower is better.

|  | Raw features | ft stage4 | Ft BN | CBN |
|---|---|---|---|---|
| Crop | 29.92% | 27.48% | 27.94% | 25.06% |
| Crop + Spatial + Category | 22.55% | 22.68% | 22.42% | **19.52**% |
| Spatial + Category | 21.5% | | | |

where $w_{box}$ and $h_{box}$ denote the width and height of the bounding box, respectively. We convert the object category $c$ into a dense category embedding using a learned look-up table. Finally, we use an LSTM to encode the current question $q$. We then concatenate all embeddings into a single vector and feed it as input to a single hidden layer MLP that outputs the final answer distribution using a softmax layer.

## 4.3  Baselines

For both datasets we empirically investigate several modifications to the described architectures. We refer to **MODERN** when we apply conditional batch normalization to all layers of ResNet-50, as described in section 3. To verify that the gains from MODERN are not coming from increased model capacity, we include two baselines with more capacity. The first model finetunes the layers of stage 4 of ResNet-50 of our baseline model. This is common practice when we transfer a pretrained network to a new task, and we refer it to as **Ft Stage 4**. We also introduce a novel baseline **Ft BN**, which consist of finetuning all $\beta$ and $\gamma$ parameters of ResNet-50, while freezing all its weights.

For VQA, we report the results of two state-of-the-art architectures, namely, Multimodal Compact Bilinear pooling network (MCB) [9] (Winner of the VQA challenge 2016) and MUTAN [2]. Both approaches employ an (approximate) bilinear pooling mechanism to fuse the language and vision embedding by respectively using a random projection and a tensor decomposition. In addition, we re-implement and run the MRN model described in Section 4.1. When benchmarking state-of-the-art models, we train on the training set, proceed early stopping on the validation set and report accuracy on the test set (test-dev in the case of VQA.)

## 4.4  Results

**VQA**   We report the best validation accuracy of the outlined methods on the VQA task in Table1. Note that we use input images of size $224x224$ when we compare MODERN against the baselines (as well as for the ablation study presented in Table 2a. Our initial baseline achieves $58.05\%$ accuracy, and we find that finetuning the last layers (*Ft Stage 4*) does not improve this performance ($56.91\%$). Interestingly, just finetuning the batch norm parameters (Ft BN) significantly improves the accuracy to $58.98\%$. We see another significant performance jump when we condition the batch normalization on the question input (*MODERN*), which improves our baseline with almost 2 accuracy points to $60.82\%$.

Because state-of-the-art models use images of size 448x448, we also include the results of the baseline architecture on these larger images. As seen in Table1, this nearly matches the state of the art results with a $62.15\%$. As MODERN does not rely on a specific attention mechanism, we then combine our proposed method with MRN [13, 14] architecture, and observe that outperforms the state-of-the-art MCB model [9] by half a point. Please note that we select MRN [13, 14] over MCB [9] as the latter requires fewer weight parameters and is more stable to train.

Note that the presented results use a ResNet-50 while other models rely on extracted image embedding from a ResNet-152. For sake of comparison, we run the baseline models with extracted image embedding from a ResNet-50. Also for the more advanced MRN architecture, we observe performance gains of approximately 2 accuracy points.

**GuessWhat?!**   We report the best test errors for the outlined method on the Oracle task of Guess-What?! in Table 3. We first compare the results when we only feed the crop of the selected object to the model. We observe the same trend as in VQA. With an error of 25.06%, CBN performs better than than either fine-tuning the final block (27.48% error) or the batch-norm parameters (27.94%

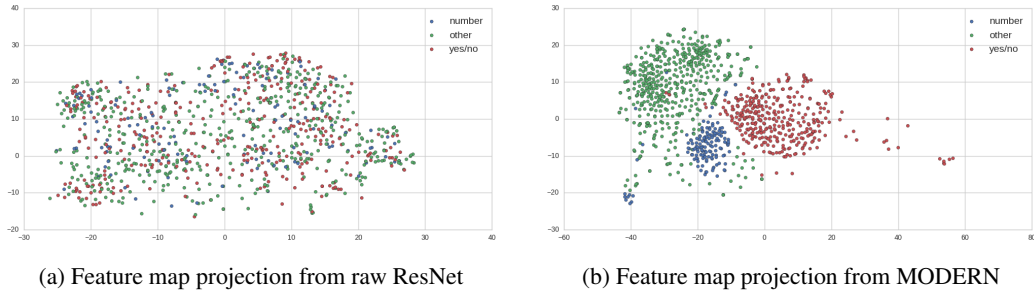

(a) Feature map projection from raw ResNet    (b) Feature map projection from MODERN

Figure 4: t-SNE projection of feature maps (before attention mechanism) of ResNet and MODERN. Points are colored according to the answer type of VQA. Whilst there are no clusters with raw features, MODERN successfully modulates the image feature towards specific answer types.

error), which in turn improve over just using the raw features (29.92% error). Note that the relative improvement (5 error points) for CBN is much bigger for GuessWhat?! than for VQA.

We therefore also investigate the performance of the methods when we include the spatial and category information. We observe that finetuning the last layers or BN parameters does not improve the performance, while MODERN improves the best reported test error with 2 points to $19.52\%$ error.

### 4.5    Discussion

By analyzing the results from both VQA and GuessWhat?! experiments, it is possible to have a better insight regarding MODERN capabilities.

**MODERN vs Fine tuning**    In both experiments, MODERN outperforms Ft BN. Both methods update the same ResNet parameters so this demonstrates that it is important to condition on the language representation. MODERN also outperforms Ft Stage 4 on both tasks which shows that the performance gain of MODERN is not due to the increased model capacity.

**Conditional embedding**    In the provided baselines of the Oracle task of GuessWhat?! [6], the authors observed that the best test error (21.5%) is obtained by only providing the object category and its spatial location. For this model, including the raw features of the object crop actually deteriorates the performance to 22.55% error. This means that this baseline fails to extract relevant information from the images which is not in the handcrafted features. Therefore the Oracle can not answer correctly questions which requires more than the use of spatial information and object category. In the baseline model, the embedding of the crop from a generic ResNet does not help even when we finetune stage 4 or BN. In contrast, applying MODERN helps to better answer questions as the test error drops by 2 points.

**Ablation study**    We investigate the impact of only modulating the top layers of a ResNet. We report these results in Table 2. Interestingly, we observe that the performance slowly decreases when we apply CBN exclusively to later stages. We stress that for best performance it's important to modulate all stages, but if computational resources are limited we recommend to apply it to the two last stages.

**Visualizing the representations**    In order to gain more insight into our proposed fusion mechanism, we compare visualizations of the visual embeddings created by our baseline model and MODERN. We first randomly picked 1000 unique image/question pairs from the *validation set* of VQA. For the trained MODERN model, we extract image features just before the attention mechanism of MODERN, which we will compare with extracted raw ResNet-50 features and finetune ResNet-50 (Block4 and batchnorm parameters). We first decrease the dimensionality by average pooling over the spatial dimensions of the feature map, and subsequently apply t-SNE [25] to these set of embeddings. We color the points according to the answer type provided by the VQA dataset, and show these visualizations for both models in Fig 4 and Fig 7 in the Appendix B. Interestingly, we observe that all answer types are spread out for raw image features and finetuned features. In contrast, the representations of MODERN are cleanly grouped into three answer types. This demonstrates that MODERN successfully disentangles the images representations by answer type which is likely to

ease the later fusion process. While finetuning models does cluster features, there is no direct link between those clusters and the answer type. These results indicate that MODERN successfully learns representation that differs from classic finetuning strategies. In Appendix B, we visualize the feature disentangling process stage by stage. It is possible to spot some sub-clusters in the t-SNE representation, as in fact they correspond to image and question pairs which are similar but not explicitly tagged in the VQA dataset. For example, in appendix B the Fig. 6 we highlight pairs where the answer is a color.

## 5 Related work

MODERN is related to a lot of recent work in VQA[1]. The majority of proposed methods use a similar computational pipeline introduced by [17, 20]. First, extract high-level image features from a ImageNet pretrained convnet, while independently processing the question using RNN. Some work has focused on the top level fusing mechanism of the language and visual vectors. For instance, it was shown that we can improve upon classic concatenation by an element-wise product [1, 13, 14], Tucker decomposition [2], bilinear pooling [9] or more exotic approaches [18]. Another line of research has investigated the role of attention mechanisms in VQA [26, 11, 28]. The authors of [11] propose a co-attention model over visual and language embeddings, while [28] proposes to stack several spatial attention mechanisms. Although an attention mechanism can be thought of as modulating the visual features by a language, we stress that such mechanism act on the high-level features. In contrast, our work modulates the visual processing from the very start.

MODERN is inspired by conditional instance normalization (CIN) [7] that was successfully applied to image style transfer. While previous methods transfered one image style per network, [7] showed that up to 32 styles could be compressed into a single network by sharing the convolutional filters and learning style-specific normalization parameters. There are notable differences with our work. First, [7] uses a non-differentiable table lookup for the normalization parameters while we propose a differentiable mapping from the question embedding. Second, we predict a change on the normalization parameters of a pretrained convolutional network while keeping the convolutional filters fixed. In CIN, all parameters, including the transposed convolutional filters, are trained. To the best of our knowledge, this is the first paper to conditionally modulate the vision processing using the normalization parameters.

## 6 Conclusion

In this paper, we introduce Conditional Batch Normalization (CBN) as a novel fusion mechanism to modulate all layers of a visual processing network. Specifically, we applied CBN to a pre-trained ResNet, leading to the proposed MODERN architecture. Our approach is motivated by recent evidence from neuroscience suggesting that language influences the early stages of visual processing. One of the strengths of MODERN is that it can be incorporated into existing architectures, and our experiments demonstrate that this significantly improves the baseline models. We also found that it is important to modulate the entire visual signal to obtain maximum performance gains.

While this paper focuses on text and images, MODERN can be extended to neural architecture dealing with other modalities such as sound or video. More broadly, CBN can could also be applied to modulate the internal representation of any deep network with respect to any embedding regardless of the underlying task. For instance, signal modulation through batch norm parameters may also be beneficial for reinforcement learning, natural language processing or adversarial training tasks.

## Acknowledgements

The authors would like to acknowledge the stimulating research environment of the SequeL lab. We thank Vincent Dumoulin for helpful discussions about conditional batch normalization. We acknowledge the following agencies for research funding and computing support: CHISTERA IGLU and CPER Nord-Pas de Calais/FEDER DATA Advanced data science and technologies 2015-2020, NSERC, Calcul Québec, Compute Canada, the Canada Research Chairs and CIFAR. We thank NVIDIA for providing access to a DGX-1 machine used in this work.

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
