[Supplementary Material]

# A Hyperparameters

In this section, we list all hyperparameters of the architectures we used. We will release our code (in TensorFlow) to replicate our experiments.

Table 4: GuessWhat?! Oracle hyperparameters

| | | |
|---|---|---|
| Question | word embedding size | 300 |
| | number of LSTM | 1 |
| | number of LSTM hidden units | 1024 |
| | use Glove | False |
| Object category | number of categories | 90 |
| | category look-up table dimension | 512 |
| Crop | crop size | 224x224x3 |
| | surrounding factor | 1.1 |
| CBN | selected blocks | all |
| | number of MLP hidden units | 512 |
| | ResNet | ResNet-50v1 |
| Fusion block | number of MLP hidden units | 512 |
| Optimizer | Name | Adam |
| | Learning rate | 1e-4 |
| | Clip value | 3 |
| | number of epoch | 10 |
| | batch size | 32 |

Table 5: VQA hyperparameters

| | | |
|---|---|---|
| Question | word embedding size | 300 |
| | number of LSTM | 2 |
| | number of LSTM hidden units | 1024 |
| | use Glove | True (dim300) |
| Image | image size | 224x224x3 |
| | attention mechanism | spatial |
| | number of units for attention | 512 |
| CBN | selected blocks | all |
| | number of MLP hidden units | 512 |
| | ResNet | ResNet-50v1 |
| Fusion block | fusion embedding size | 1024 |
| | number of MLP hidden units | 512 |
| | number of answers | 2000 |
| Optimizer | Name | Adam |
| | Learning rate | 2e-4 |
| | Clip value | 5 |
| | number of epoch | 20 |
| | batch size | 32 |

# B T-SNE visualization

(a) Feature map projection from MODERN (Stage4)

(b) Feature map projection from MODERN (Stage3)

(c) Feature map projection from MODERN (Stage2)

(d) Feature map projection from MODERN (Stage1)

(a) Feature map projection from raw ResNet

(b) Feature map projection from MODERN

Figure 6: t-SNE projection of feature maps of Reset and MODERN by coloring. Points are colored according to the question type (here, colors) of the image/question pair from the VQA dataset.

(a) Feature map projection from ResNet + Block4 Ft

(b) Feature map projection from ResNet + BatchNorm ft

Figure 7: t-SNE projection of feature maps (before attention mechanism) of finetune ResNet. Points are colored according to the answer type of VQA. No answer-type clusters can be observed in both cases.