[Reviews · NeurIPS 2017]

Reviewer 1



Overall Impression: I think this paper introduces a novel and interesting idea that is likely to spark future experimentation towards multi-modal early-fusion methods. However, the presentation and the writing could use additional attention. The experiments demonstrate the effectiveness of the approach on multiple tasks though they are a bit narrow to justify the proposed method outside of the application domain of vision + language. I think further iterations on the text and additional experiments with other model architectures or different types of multi-modal data would strengthen this submission. Strengths: + I like the neurological motivations for the CBN approach and appreciate its simplicity. + Comparing fine-tuning batch norm parameters (Ft BN) vs the question-conditioned batch norm predictions provided an interesting ablation. It seems like adjusting to the new image statistics (Ft BN) results in significant improvement (~1% VQA, 2% Crop GuessWhich) which is then doubled by conditioned on question (~2% VQA, 4% Crop GuessWhich). + I appreciate the promise of public code to reproduce the experimental results. + The tSNE plots are quite interesting and show that the language conditional modulation seems to have a significant effect on the visual features. Weaknesses: - I don't understand why Section 2.1 is included. Batch Normalization is a general technique as is the proposed Conditional Batch Normalization (CBN). The description of the proposed methodology seems independent of the choice of model and the time spent describing the ResNet architecture could be better used to provide greater motivation and intuition for the proposed CBN approach. - On that note, I understand the neurological motivation for why early vision may benefit from language modulation, but the argument for why this should be done through the normalization parameters is less well argued (especially in Section 3). The intro mentions the proposed approach reduces over-fitting compared to fine-tuning but doesn't discuss CBN in the context of alternative early-fusion strategies. - As CBN is a general method, I would have been more convinced by improvements in performance across multiple model architectures for vision + language tasks. For instance, CBN seems directly applicable to the MCB architecture. I acknowledge that needing to backprop through the CNN causes memory concerns which might be limiting. - Given the argument for early modulation of vision, it is a bit surprising that applying CBN to Stage 4 (the highest level stage) accounts for majority of the improvement in both the VQA and GuessWhat tasks. Some added discussion in this section might be useful. The supplementary figures are also interesting, showing that question conditioned separations in image space only occur after later stages. - Figures 2 and 3 seem somewhat redundant. Minor things: - I would have liked to see how different questions change the feature representation of a single image. Perhaps by applying some gradient visualization method to the visual features when changing the question? - Consider adding a space before citation brackets. - Bolding of the baseline models is inconsistent. - Eq 2 has a gamma_j rather than gamma_c L34 'to let the question to attend' -> 'to let the question attend' L42 missing citation L53 first discussion of batch norm missing citation L58 "to which we refer as" -> "which we refer to as" L89 "is achieved a" -> "is achieved through a"

Reviewer 2



The paper proposes a novel method called conditional batch normalization (CBN) to be applied on top of existing visual question answering models in order to modulate the visual processing with language information from the question in the early stages. In the proposed method, only the parameters of the batch norm layer of a pre-trained CNN are updated with the VQA loss by conditioning them on the LSTM embedding of the input question. The paper evaluates the effectiveness of CBN on two VQA datasets – the VQA dataset from Antol et al., ICCV15 and the GuessWhat?! dataset from Vries et al., CVPR17. The experimental results show that CBN helps improve the performance on VQA by significant amount. The paper also studies the effectiveness of adding CBN to different layers and shows that adding CBN to last (top) 2 layers of CNN helps the most. The paper also shows quantitatively that the improvements in VQA performance are not merely due to fine-tuning of CNN by showing that the proposed model performs better than a model in which the Batch Norm parameters are fine-tuned but without conditioning on the language. Hence demonstrating that modulating with language helps. Strengths: 1. The paper is well-motivated and the idea of modulating early visual processing by language is novel and interesting for VQA task. 2. The proposed contribution (CBN) can be added on top of any existing VQA model, hence making it widely applicable. 3. The ablation studies are meaningful and are informative about how much of early modulation by language helps. 4. The paper provides the details of the hyper-parameters, hence making the work reproducible. Weaknesses: 1. The main contribution of the paper is CBN. But the experimental results in the paper are not advancing the state-of-art in VQA (on the VQA dataset which has been out for a while and a lot of advancement has been made on this dataset), perhaps because the VQA model used in the paper on top of which CBN is applied is not the best one out there. But in order to claim that CBN should help even the more powerful VQA models, I would like the authors to conduct experiments on more than one VQA model – favorably the ones which are closer to state-of-art (and whose codes are publicly available) such as MCB (Fukui et al., EMNLP16), HieCoAtt (Lu et al., NIPS16). It could be the case that these more powerful VQA models are already so powerful that the proposed early modulating does not help. So, it is good to know if the proposed conditional batch norm can advance the state-of-art in VQA or not. 2. L170: it would be good to know how much of performance difference this (using different image sizes and different variations of ResNets) can lead to? 3. In table 1, the results on the VQA dataset are reported on the test-dev split. However, as mentioned in the guidelines from the VQA dataset authors (http://www.visualqa.org/vqa_v1_challenge.html), numbers should be reported on test-standard split because one can overfit to test-dev split by uploading multiple entries. 4. Table 2, applying Conditional Batch Norm to layer 2 in addition to layers 3 and 4 deteriorates performance for GuessWhat?! compared to when CBN is applied to layers 4 and 3 only. Could authors please throw some light on this? Why do they think this might be happening? 5. Figure 4 visualization: the visualization in figure (a) is from ResNet which is not finetuned at all. So, it is not very surprising to see that there are not clear clusters for answer types. However, the visualization in figure (b) is using ResNet whose batch norm parameters have been finetuned with question information. So, I think a more meaningful comparison of figure (b) would be with the visualization from Ft BN ResNet in figure (a). 6. The first two bullets about contributions (at the end of the intro) can be combined together. 7. Other errors/typos: a. L14 and 15: repetition of word “imagine” b. L42: missing reference c. L56: impact -> impacts Post-rebuttal comments: The new results of applying CBN on the MRN model are interesting and convincing that CBN helps fairly developed VQA models as well (the results have not been reported on state-of-art VQA model). So, I would like to recommend acceptance of the paper. However I still have few comments -- 1. It seems that there is still some confusion about test-standard and test-dev splits of the VQA dataset. In the rebuttal, the authors report the performance of the MCB model to be 62.5% on test-standard split. However, 62.5% seems to be the performance of the MCB model on the test-dev split as per table 1 in the MCB paper (https://arxiv.org/pdf/1606.01847.pdf). 2. The reproduced performance reported on MRN model seems close to that reported in the MRN paper when the model is trained using VQA train + val data. I would like the authors to clarify in the final version if they used train + val or just train to train the MRN and MRN + CBN models. And if train + val is being used, the performance can't be compared with 62.5% of MCB because that is when MCB is trained on train only. When MCB is trained on train + val, the performance is around 64% (table 4 in MCB paper). 3. The citation for the MRN model (in the rebuttal) is incorrect. It should be -- @inproceedings{kim2016multimodal, title={Multimodal residual learning for visual qa}, author={Kim, Jin-Hwa and Lee, Sang-Woo and Kwak, Donghyun and Heo, Min-Oh and Kim, Jeonghee and Ha, Jung-Woo and Zhang, Byoung-Tak}, booktitle={Advances in Neural Information Processing Systems}, pages={361--369}, year={2016} } 4. As AR2 and AR3, I would be interested in seeing if the findings from ResNet carry over to other CNN architectures such as VGGNet as well.

Reviewer 3



Summary: The current work takes a different approach to fusing modalities for tasks in the intersection of language and vision, than most of the other works. Instead of fusing these modalities after extracting independent representations, the authors aim to modulate the visual features using the text. In particular, they aim to learn the batch normalization parameters of ResNet, which is typically used to extract visual representation, conditioned on the text. They call such it ‘Conditional Batch Normalization’ (CBN) and the resultant network as ModRNet. Experiments on two tasks shows the effectiveness of the proposed approach. Strengths: (a) Fusing both the modalities as early as the visual feature extraction is novel idea, to the best of my knowledge. This is well motivated by the paper, drawing parallels to findings in neuroscience on how language can influence the response to visual stimulus in human brain. (b) Batch normalization parameters as a means to fuse modalities is a good choice as it scales only with the number of channels, thus greatly limiting the number of additional parameters to be learnt. Further, other parameters of the network being frozen exemplifies the simplicity of the approach. (c) The paper contains good ablation studies along with analyzing finetuning ResNet on the task vs ModRNet. I thoroughly enjoyed the experiments and the discussions that followed. Weaknesses: (a) Experiments in the paper are limited to ResNet architecture for feature extraction. Though not a strict weakness, one does wonder about the generalizability of CBN on other networks, perhaps batch normalized version of VGG? Evidence that this is somehow a property of ResNet, perhaps due to the residual connection, would also suffice. (b) Highly recommend the authors to carefully read the manuscript and resolve typos/grammatical issues that hinder the reading and understanding of the proposed approach. Few of the errors are listed towards the end. Comments: (a) L13-16 sentences don’t flow well. While the initial sentences suggest our capability to model such tasks, the later sentences state that these remain a long-standing challenge. (b) L185 - The task of oracle has not been described (with the input - outputs clearly mentioned) till this point. This makes the understanding of specifics that follow difficult. (c) L212 - Retrain their model with the same visual features on same sized images as the current work, for perfect comparison ? Typos: (a) L4 - inputs (b) L17 - towards (c) L42 - Missing reference or a typo? (d) L150 - latex typos (e) L190 - ReLU (f) L218, L267 - typos Post discussion: The authors seemed to have mostly done a good job with the rebuttal. After looking at other reviewer's comments and the rebuttal, I continue to feel that the paper has novel contributions bringing new insights which future works can build on. However, I am still curious about the generalization to other networks like my other reviewers. Showing some evidence on other networks (batch normalized version of VGG) would make the paper stronger, even if the findings suggest that it is not useful for other architectures. The authors do mention the lack of pretrained batch norm VGG models on tensorflow, but would be great if they could use pytorch pretrained models in time for the camera ready.